# Neurophysiological Mechanisms Underlying Cortical Hyper-Excitability in Amyotrophic Lateral Sclerosis: A Review

**DOI:** 10.3390/brainsci11050549

**Published:** 2021-04-27

**Authors:** Jonu Pradhan, Mark C. Bellingham

**Affiliations:** Faculty of Medicine, School of Biomedical Sciences, The University of Queensland, Brisbane, QLD 4072, Australia; j.pradhan@uq.edu.au

**Keywords:** amyotrophic lateral sclerosis, electrophysiology, motor cortex, upper motor neuron, synaptic transmission, glutamate, neuronal structure

## Abstract

Amyotrophic lateral sclerosis (ALS) is a progressive neuromotor disease characterized by the loss of upper and lower motor neurons (MNs), resulting in muscle paralysis and death. Early cortical hyper-excitability is a common pathological process observed clinically and in animal disease models. Although the mechanisms that underlie cortical hyper-excitability are not completely understood, the molecular and cellular mechanisms that cause enhanced neuronal intrinsic excitability and changes in excitatory and inhibitory synaptic activity are starting to emerge. Here, we review the evidence for an anterograde glutamatergic excitotoxic process, leading to cortical hyper-excitability via intrinsic cellular and synaptic mechanisms and for the role of interneurons in establishing disinhibition in clinical and experimental settings. Understanding the mechanisms that lead to these complex pathological processes will likely produce key insights towards developing novel therapeutic strategies to rescue upper MNs, thus alleviating the impact of this fatal disease.

## 1. Introduction

First described by Jean-Martin Charcot in 1869, amyotrophic lateral sclerosis (ALS), one of the most common neuromotor diseases, is characterized by an inexorable loss of upper and lower motor neurons (MNs). The upper motor neurons of the primary motor cortex are large pyramidal neurons in layer V (LVPNs), whose axonal projections form the corticospinal tracts, which descend to directly or indirectly excite the lower motor neurons (LMNs) of brainstem and the spinal cord [1,2]. Although lower MNs are the final output pathway that integrate motor output and regulate muscle activity, the presence of corticospinal monosynaptic or polysynaptic axonal projections that originate in the primary motor cortex, dorsal prefrontal cortex and somatosensory cortex [3,4] and relay cortical output to the LMNs allow fine regulation of neuromotor execution in humans [5]. There is substantial evidence, both clinically and in animal models, suggesting significant cortical disruption early in ALS. However, what makes the upper MNs vulnerable to degeneration and how this perturbs the neuromotor circuitry is unclear.

The clinical outcome in ALS is largely based on lower MN degeneration, presenting mostly as the onset of weakness in limb (80%) or bulbar (20%) muscles [6]. A relentless progression occurs, with muscle weakness spreading to multiple muscles, paralysis and death from respiratory failure within two to three years of diagnosis in 50% of ALS patients [7]. Cognitive abnormalities are increasingly recognized in ALS patients, with fronto-temporal dementia in about 5 to 15% ALS cases, worsening the prognosis and disease progression [8,9]. Classically, a clear family history is present in only 10% of all ALS cases and has been increasingly (but not always) associated with a genetic cause, while the remaining 90% have no family history and are classified as sporadic ALS (sALS), although a genetic cause can be identified in about 10% of these patients [2,10,11]. This makes ALS a disease of complex pathology and the continued discovery of new genetic causes highlights this.

In an attempt to identify the origin of neurodegeneration and the correlation between upper MN and lower MN dysfunction, the “dying forward” [5] hypothesis was conceptualized. One proposed pathogenic mechanism that culminates in the death of upper and lower MNs is glutamate-induced excitotoxicity which results from either excessive presynaptic glutamate release or defective glutamate reuptake. This depolarizes the post-synaptic neuron and increases calcium influx, thus leading to hyper-excitability [12,13]. Hyper-excitability is a commonly observed feature of different cell types at several locations, including LVPNs [14,15] brainstem motor neurons [16], spinal motor neurons [17,18] and skeletal muscles [19] that may contribute to pathology in ALS. Cortical hyper-excitability seems to be one of the most common features preceding the degeneration of upper MNs [14,15,20] and LMNs [21,22]. This review concentrates on the neurophysiology of hyper-excitability in upper MNs and the probable mechanisms involved.

## 2. Excitability and Hyper-Excitability in ALS: General Concepts

Excitability of a neuron is a measure of its electrophysiological properties. This relates to the neuron’s ability to produce an output or action potential (AP) by depolarizing its membrane potential to a threshold level in response to an input. This intrinsic ability of a neuron is referred to as its “intrinsic excitability” and is defined by factors including the types and number of excitatory receptors and voltage gated ion channels (e.g., Na^+^, K^+^) present, which shapes the neuronal output [12]. Similarly, neuronal activity typically refers to the frequency of spontaneously elicited APs or post-synaptic currents, and this depends on the neuron’s intrinsic excitability, its synaptic input strength and excitation/inhibition balance, and the coordination of excitatory and inhibitory synaptic input to generate an AP.

The concept of functional hyper-excitability or exaggerated neuronal output in ALS involving cortical dysfunction has been described by many [23,24] and is considered a hallmark feature which may contribute to an excitotoxic environment via altered intrinsic neuronal properties or synaptic mechanisms [14,15,25,26]. Clinically, hyper-excitability in upper MNs is directly associated with corticospinal tract abnormality and is characterized by the presence of enhanced muscle tone and increased reflexes, including brisk reflexes and spasticity [27,28], symptoms that are considered clinical hallmarks of cortical involvement. Moreover, cortical dysfunction and hyper-excitability are now considered as early diagnostic signs of ALS [29].

### 2.1. Hyper-Excitability in Upper MNs: Influence on Cortical Dysfunction

Eisen and colleagues first emphasized the direct cortical involvement in ALS with their dying forward hypothesis, whereby the primary cortical dysfunction in upper MNs triggers the subsequent pathologic degeneration and death of lower MNs [5,29]. A key part of this hypothesis was the evidence that lower MNs in Onuf’s, oculomotor and abducens nuclei, which are spared or mildly affected late in ALS, lack direct monosynaptic neuronal projections in humans, indicating a role for direct cortical inputs in relaying cortical dysfunction [5]. Additionally, cortical hyper-excitability is observed prior to the symptomatic stage clinically [21,22,30] and in experimental animal models [14,15,20,31], clearly underlining a role for anterograde cortical dysfunction in ALS. Importantly, recent advances in clinical, molecular, neuroimaging and genetic studies further emphasize the significant pathological influence of hyper-excitability-induced upper MN dysfunction in ALS [29,32]. Moreover, cortical hyper-excitability-associated excitotoxicity is a result of enhanced intrinsic properties and synaptic dysfunction [33].

A counter to the original dying forward hypothesis is that direct monosynaptic contacts between upper and lower MNs are thought to direct fine voluntary motor control and are found in humans and some primates [34], although both mono- and polysynaptic connections co-exist in the same species [35,36]. However, despite a lack of direct corticospinal inputs onto lower MNs, rodent models of ALS show prominent upper MN degeneration preceding lower MN loss [4,37,38], while selective knockdown of ALS gene mutations in rodent cortex delays disease progress and lower MN loss and extends survival [39,40]. As corticospinal influence is relayed to lower MNs by local oligosynaptic excitatory interneuronal networks in non-primate species [41,42], it seems likely that this circuitry can still mediate a dying forward influence. Hence, despite variation in the morphological and functional properties of these neurons between species, lower MNs are directly or indirectly modulated by upper MNs via corticospinal descending projections from the motor cortex [41,42,43,44] and this influence from upper MNs are deleterious for lower MN survival in ALS [39,40]. A recent clinical study in ALS onset patients showing absence of hyper-excitability in spinal MNs [45] could possibly reflect a compensatory mechanism in the surviving MNs, given that not all lower MNs are hyper-excitable and their excitability status correlates with their survival [46,47,48]. Upper MN degeneration precedes lower MN loss [4,37,38], clearly implying the possibility of a dying forward influence in ALS.

#### 2.1.1. Intrinsic Excitability-Mediated Cortical Dysfunction

The intrinsic excitability of a neuron reflects its electrical properties, comprising the membrane density and distribution of ionic conductances and receptors present on its particular dendritic morphology. The maintenance of appropriate functional output is strongly influenced by the number and properties of synaptic inputs [49]. Increased depolarizing currents are reported to result in an enhanced firing rate [49,50], and hence, it is highly likely that increased intrinsic excitability and excitotoxic susceptibility in ALS are linked [20].

One prime factor affecting the neuronal excitability is the persistent inward current (PIC) generated by a inactivation-resistant population of Na^+^ channels, referred to as the persistent sodium current (INaP) [51]. In ALS, the intrinsic excitability of cortical MNs are positively skewed by the enhanced current density of INaP, increasing the firing frequency triggered by elevated Na^+^ current and depleted K^+^ conductance [20] and producing persistent depolarizations due to rapid activation and slow decay kinetics [52,53]. Several other studies reinforce this finding, including increased AP firing observed in layer V pyramidal neurons [15] and in motor cortex slices [54]. More recently, the NaV1.6 channel localized in the initial segment of the axon have been shown to be overexpressed in cortical MNs in ALS [55], to the point where AP firing initiates [56], clearly confirming sustained depolarization-induced hyper-excitability in dying cortical MNs [20]. A depolarizing stimulus will activate any PIC present to generate AP firing in MNs [50], and hence, elevated PIC observed in ALS can contribute to hyper-excitability-induced excitotoxicity [20] by directly enhancing intracellular Ca^2+^ influx and MN firing frequency [57,58]. Furthermore, the selective suppression of the INaP and firing frequency in LMNs in animal models of ALS [16,59,60], and in clinical studies of cortical hyper-excitability [61] by riluzole, a glutamatergic antagonist and inhibitor of Na^+^ channel activity, further reinforces the correlation between increased PIC and cortical dysfunction [20,54,61].

Similarly, another contributor to MN hyper-excitability in ALS is reduced K^+^ conductance, documented in clinical studies showing peripheral motor axonal excitability [62,63] and in patient-derived MNs showing reduced amplitude of delayed rectifier K^+^ currents [64]. However, whether the corticomotor neurons exhibit reduced K^+^ conductance remains to be explored. Activating Kv7.2/3 potassium channels (also known as the M current) by retigabine, an activator of Kv7.2/3 potassium channels drives K^+^ efflux, counterbalancing the Na^+^ influx and thus dampening MN hyper-excitability as observed in ALS patient-derived MNs [64] and LVPNs of mouse models [65]. This evidence supports the idea that intrinsic neuronal properties resulting in hyper-excitability may be a combination of increased Na^+^ and decreased K^+^ conductances, providing additional avenues for therapeutic exploration.

#### 2.1.2. Synaptic Excitability-Mediated Cortical Dysfunction

Cortical hyper-excitability in ALS is also associated with both functional and morphological synaptic changes [14,15,31,66]. The synaptic regulation of neuronal activity is determined via the number, distribution and timing of excitatory and inhibitory neurotransmission. A major part of synaptic dysfunction in ALS is increased glutamatergic excitatory neurotransmission, giving rise to excessive glutamate, a key driver of excitotoxicity [67,68]. During excitatory neurotransmission, glutamate is secreted from the pre-synaptic neuron and acts to rapidly activate AMPA and NMDA receptors [68]. Extracellular glutamate levels are regulated within the CNS via interactive functions, namely uptake and synthesis through glutamate transporters and the levels of excitatory and inhibitory neurotransmission [68]. Taking into account the negative effects of glutamate reported on neurons [68], the enhanced glutamate observed in ALS patients provides a clear correlation between increased glutamate levels, hyper-excitability and excitotoxicity. Enhanced glutamate mediated excitotoxicity in ALS has been shown to be a combination of excessive pre-synaptic release [67,69] and inhibited uptake from the synaptic cleft due to ineffective astroglial glutamate transporters (EAAT2) observed in the spinal cord [70] and motor cortex of affected ALS patients [71,72] and in rodent models of ALS [73,74,75], resulting in glutamate retention and persistent enhanced post-synaptic NMDA and AMPA receptor activation [67,68], causing an increased intracellular influx of Ca^2+^ and Na^+^ ions resulting in a synaptic hyper-active state [69]. Additionally, the Ca^2+^ buffering capacity of MNs in ALS is reported to be impaired, with defective Ca^2+^ ATPase and Ca^2+^/Na^+^ exchangers exacerbating the increased intracellular Ca^2+^ burden [76] leading to Ca^2+^ overabundance and sustained depolarization [58,76,77]. Glutamatergic hyper-excitation is observed as the enhanced spontaneous excitatory post-synaptic current (sEPSC) frequency of LVPNs [14,15]. This cascade of events in cortical hyper-excitability and dysfunction evolves into neuronal degeneration, as mitochondrial defects cause oxidative stress-generating reactive oxygen species, eventually leading to the death of the MN [13].

#### 2.1.3. Clinical Evidence of Functional and Structural Cortical Hyper-Excitability

Clinical observation of functional and structural cortical excitatory measurements in ALS patients serve as a marker of altered cortical output function, i.e., the cortical state of excitability. Enhanced cortical excitability prior to symptom onset due to significant lower MN degeneration has been observed in ALS patients by utilizing advanced clinical neurophysiological and neuroimaging techniques [29], suggesting that cortical dysfunction may be a significant early diagnostic marker in ALS [29,78]. Trans-cranial magnetic stimulation, an effective tool for evaluating functional cortical excitability, demonstrated a decrease in short-interval intracortical inhibition (a measure of cortical inhibitory function), increased cortico-conduction time and enhanced intracortical facilitation in Betz cells (layer V output neurons) in ALS patients [21,22,78,79,80]. Short interval intercortical inhibition (SICI) is an activation of a subthreshold inhibitory circuit via subthreshold stimulus, which represents an equilibrium between the robust inhibitory effect mediated by GABAergic interneurons in conjunction with weaker cortical glutamatergic facilitatory effects from pyramidal neurons. This process increases the threshold for the generation of an evoked response; hence, the observation of low threshold elicitation of a motor-evoked potential (MEP) indicates cortical hyper-excitability [22,24,81]. Given that synaptic transmission is a balance between inhibitory and excitatory transmission, the presence of reduced SICI, reflecting inhibitory interneuronal dysfunction [82,83], alongside increased MEP amplitude, indicating enhanced excitatory transmission mediated via glutamate [84], clearly signals that cortical hyperexcitability is most likely a combination of cortical disinhibition and enhanced excitation. Furthermore, functional MRI studies provide additional evidence for early cortical structural dysfunction, demonstrating cortical thinning of the primary motor cortex and temporal cortex [85,86]. Likewise, diffusion tensor imaging showed abnormalities in the corticospinal tracts projecting towards the lower MNs in ALS patients [87,88,89,90], suggesting the loss of cortical MNs and degeneration of their corticospinal projections. Additionally, the partial reduction of cortical hyper-excitability in ALS patients by riluzole administration further reinforces the pathogenic action of cortical hyper-excitability in ALS [61,91].

## 3. Morphological Changes Correlated with Cortical Hyper-Excitability

Corticospinal neurons, namely LVPNs in the deep motor cortical layer, are the source of corticospinal projections to spinal cord controlling voluntary movement execution. Besides direct corticospinal projections, LVPNs also make direct corticofugal connections to the thalamus, cortico-cortical connections to layer 4 and layer 2/3 motor cortex, and corticobulbar connections to motor nuclei in the brainstem [92]. Evidence of early upper MN degeneration has been demonstrated in various rodent ALS models [4,37,93,94] and more subtle changes in LVPN morphology have been observed as an early loss of dendritic spines and shrinkage of apical dendrites [14,66,95]. Additionally, in ALS patients, Betz cells of the motor cortex (the largest LVPNs) also display apical dendrite degeneration with intensive vacuolation, disintegration of their dendritic architecture and reduced synapse number, indicative of functional changes in cortical neurons [96].

## 4. The Role of Inhibitory Cortical Interneurons in Cortical Hyper-Excitability

The maintenance of normal neuronal activity in the cortical neural circuit is an interaction between the excitatory pyramidal neurons and abundant interneurons that mainly cause inhibition via γ-aminobutyric acid (GABA) neurotransmission. GABAergic interneurons in the motor cortex can be activated via a sub-threshold stimulus [97]. In ALS patients, dysfunction of cortical inhibitory interneuronal activity is observed as reduced or absent SICI mediated by cortical GABAergic interneurons [22,78,81], alongside increased cortico-neuronal excitability; this clearly implicates an impaired inhibition/excitation balance induction of cortical hyper-excitability. Additionally, reduced levels of GABA in the motor cortex [98], decreased mRNA expression of the alpha subunit of GABA_A_ receptors [99] and functional neuroimaging studies revealing decreased binding of flumazenil, a GABA_A_ receptor ligand, in sporadic ALS patients [100] provide direct evidence that decreased inhibition may contribute to cortical hyper-excitability. Furthermore, histological studies have shown reduced parvalbumin interneuron numbers in post-mortem motor cortices of ALS patients [101] and animal models of ALS [102,103], and a decreased number of neuropeptide Y-positive interneurons at symptom onset coupled with progressive reduction of calretinin-positive interneurons from symptom onset up to the end stage in the motor cortex of an ALS mouse model [104]. A more recent electrophysiological study revealed decreased intracortical inhibition in layer V interneurons of motor cortex and a rescue of this disinhibition-induced hyper-excitability by increasing interneuronal activity [102]. Additionally, cortical parvalbumin interneurons [105] and somatostatin interneurons [106] have also been shown to display increased intrinsic and synaptic excitability. Given that cortical interneurons can display either hyper-excitability [105,106] or hypo-excitability [102], it remains to be determined whether cortical hyper-excitability in a rodent model occurs in addition to, or as a consequence of, the hyper-excitability of cortical excitatory and inhibitory neuronal populations. Taken together, these findings clearly imply a role for interneuron dysfunction causing an imbalance in cortical excitation/inhibition and thus hyper-excitability in ALS.

## 5. Conclusions

Hyper-excitability is a prevalent feature observed clinically in ALS patients and in animal models for ALS. What causes cortical hyper-excitability and how hyper-excitability drives upper MN degeneration and death is just beginning to be unraveled. Evidence from electrophysiological and histological studies in humans and in animal models provides insight into an early cortical involvement in the disease process that precedes lower MN degeneration. Cortical hyper-excitability is clearly a result of increased excitation and impaired inhibition, and the evidence so far points to a number of molecular and cellular mechanisms that contribute to cortical hyper-excitability and that may drive the degeneration of upper MNs. It will be critically important to further characterize these mechanisms and understand their consequences, in order to develop rational therapeutic strategies that normalize hyper-excitation and alleviate the disease.

## Data Availability

Not applicable.

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
