# Peer review of "Neurophysiological Mechanisms Underlying Cortical Hyper-Excitability in Amyotrophic Lateral Sclerosis: A Review"

_brainsci, 2021, doi:10.3390/brainsci11050549_

Round 1

Reviewer 1 Report

The review by J. Pradhan and M.C. Bellingham is altogether timed, well written and informative. It could nevertheless be improved, by the addition of recent citations and the discussion of key points such as the difference between cortical hyperexcitability and cortical neuron hyperexcitability, and their consequences on LMN excitability and survival in humans and mice, given the differences of the corticospinal networks. More precisely :

1) The review suggests that the connections between  human upper and lower MN are only monosynaptic, as opposed to the rodents where those are only polysynaptic. This statement should be corrected, given that  both mono- and polysynaptic connections exist in humans and higher primates (Welniarz et al., Dev Neurobiol 2016; Lemon et al., Ann Rev Neurosci 2008).

2) Together with initial work from Thomsen and collaborators, a recent article brought experimental evidence of the corticofugal/dying forward hypothesis in ALS using mouse genetics (Burg et al., Ann Neurol 2020).

3) Additional mouse models present with presymptomatic UMN loss that are worth citing : SOD1G93A/G1H  (Zang and Cheema, Neuroscience Letters 2002) and SOD1G86R – with a focus on somatotopic relationship between loss of UMN and LMN (Marques et al. Brain Sciences 2021).

4) The authors should remain careful in comparing UMN and LMN hyperexcitability and discuss how or whether the data from LMN could be relevant to UMN, given that i) these are not the same type of neurons, ii) not all LMN present with hyperexcitability and their excitability status correlates with their survival (Delestrée et al., J Physiol 2014; Leroy et al., eLIFE 2014; Martínez-Silva et al. eLIFE 2018), iii)  UMN are upstream of LMN (directly or indirectly). The authors should thus discuss how UMN hyperexcitability may influence LMN excitability and survival, especially in mice where connections are exclusively polysynaptic and where LMN excitability varies with subtypes and time.

5) The author should probably further discuss and compare the concepts of cortical hyperexcitability, as revealed by paired-pulse TMP in patients, and neuronal hyperexcitability, as revealed by patch-clamp in rodents. Given that cortical interneurons were also reported to display either hyperexcitability (Zhang et al., Nat Neuro 2016; Kim et al. J Neurosci 2017) or hypoexcitability (Khademullah et al, Brain 2020), it remains to be determined whether rodent models of ALS recapitulate cortical hyperexcitability (in addition or as a consequence to hyperexcitability of cortical excitatory and inhibitory neuronal subpopulations).

Minor points:

Line 164 : “provide” should be “provides”

Line 211 : “corticalspinal” sould be “corticospinal”.

Author Response

Reviewer 1

The review by J. Pradhan and M.C. Bellingham is altogether timed, well written and informative. It could nevertheless be improved, by the addition of recent citations and the discussion of key points such as the difference between cortical hyperexcitability and cortical neuron hyperexcitability, and their consequences on LMN excitability and survival in humans and mice, given the differences of the corticospinal networks. More precisely:

1) The review suggests that the connections between human upper and lower MN are only monosynaptic, as opposed to the rodents where those are only polysynaptic. This statement should be corrected, given that both mono- and polysynaptic connections exist in humans and higher primates (Welniarz et al., Dev Neurobiol 2016; Lemon et al., Ann Rev Neurosci 2008).

Response 1

We thank our reviewer for pointing out the presence of both monosynaptic and polysynaptic projections between upper and lower motor neurons. We have incorporated this modification and also included the references in our revised manuscript. (Page 1, line 32; Page 3, line 107).

2) Together with initial work from Thomsen and collaborators, a recent article brought experimental evidence of the corticofugal/dying forward hypothesis in ALS using mouse genetics (Burg et al., Ann Neurol 2020).

Response 2

We thank our reviewer for adding a recent reference in addition to the initial work by Thomson et al., on evidence of the dying forward hypothesis. We have therefore added the recent work by Burg et al., Ann Neurol 2020 to our paper. (Page 3, line 114; Page 3, line 121).

3) Additional mouse models present with presymptomatic UMN loss that are worth citing: SOD1G93A/G1H (Zang and Cheema, Neuroscience Letters 2002) and SOD1G86R – with a focus on somatotopic relationship between loss of UMN and LMN (Marques et al. Brain Sciences 2021).

Response 3

We thank our reviewer for the additional reference citing the pre-symptomatic upper MN loss preceding lower MN loss. We have added SOD1G93A/G1H (Zang and Cheema, Neuroscience Letters 2002) and SOD1G86R – with a focus on somatotopic relationship between loss of UMN and LMN (Marques et al. Brain Sciences 2021) to our reference. (Page 3, line 112).

4) The authors should remain careful in comparing UMN and LMN hyperexcitability and discuss how or whether the data from LMN could be relevant to UMN, given that i) these are not the same type of neurons, ii) not all LMN present with hyperexcitability and their excitability status correlates with their survival (Delestrée et al., J Physiol 2014; Leroy et al., eLIFE 2014; Martínez-Silva et al. eLIFE 2018), iii) UMN are upstream of LMN (directly or indirectly). The authors should thus discuss how UMN hyperexcitability may influence LMN excitability and survival, especially in mice where connections are exclusively polysynaptic and where LMN excitability varies with subtypes and time.

Response 4

We thank our reviewer for detailing the careful comparison of upper and lower MN hyper-excitability and its correlation. In the revised manuscript, we have incorporated the information and references relating the upper MN hyper-excitability and the direct or indirect influence on lower MNs where LMN excitability varies with subtypes and time. (Page 3, line 118-126; Page 3, line 124).

5) The author should probably further discuss and compare the concepts of cortical hyperexcitability, as revealed by paired-pulse TMP in patients, and neuronal hyperexcitability, as revealed by patch-clamp in rodents. Given that cortical interneurons were also reported to display either hyperexcitability (Zhang et al., Nat Neuro 2016; Kim et al. J Neurosci 2017) or hypoexcitability (Khademullah et al, Brain 2020), it remains to be determined whether rodent models of ALS recapitulate cortical hyperexcitability (in addition or as a consequence to hyperexcitability of cortical excitatory and inhibitory neuronal subpopulations).

Response 5

We thank our reviewer for their insightful comment to further discuss and compare clinical cortical hyper-excitability by TMS and neuronal hyper-excitability which we have discussed further in section 2.1.3. (Page 5, line 222-232; Page 3, line 124).

Also, we have included the additional reference and the possibility that cortical hyperexcitability observed could be heralded in addition or as a consequence to hyperexcitability of cortical excitatory and inhibitory neuronal subpopulations, (Zhang et al., Nat Neuro 2016; Kim et al. J Neurosci 2017) or hypoexcitability (Khademullah et al, Brain 2020). (Page 6, line 279-284).

Minor points:

Line 164 : “provide” should be “provides”

Response: Changed accordingly, (Page 4, line 186).

Line 211 : “corticalspinal” sould be “corticospinal”.

Response: Changed accordingly, (Page 5, line 243).

Reviewer 2 Report

The authors propose a short review on hyperexcitbility in ALS. They decided to focus on cortical excitability only, missing the possible concomitant lower motor neuron hyperexcitability, clinically suggested by the presence of fasciculations early on the disease progression. In addition to this line, the authors wanted to defend the corticomotoneuronal hypotheis for ALS, in their contribution. This partial view did not consider the cons against this theory, which reduced the impact of their contribution. I would propose the authors should just keep their attention on the hyperexcitability topic.

A number of sentences are questionable. In introduction, indeed most axons present in the corticospinal tract have their origin outside the primary motor area, and the monosynaptic pathway is most related to hand discrete movements. In many familial cases no known genetic mutation is identified, but in about 10% sporadic case a genetic cause can be found.

Regarding Excitability section, spasticity and brisk reflexes derive from corticospinal tract lesion (like as stroke and spinal cord trauma) and not from cortical hyperexcitability. Really, cortical hyperexcitability causes epilepsy, which does not affect ALS patients. Both oculomotor nuclei and anal sphincter are (mildly) affected in ALS patients. Moreover, the clinical evidence for potassium channel dysfunction arrived from investigating peripheral axons (threshold tracking technique) and not from cortical studies (in humans). The way the authors approach these issues is misleading for readers. The same hold true for EAAT2 astroglial receptor dysfunction, demonstrated in the spinal cord from affected ALS patients (Rothstein et al).

Finally, early cortical hyperexcitability does not prove it antedates lower motor neuron changes. This is fashionable but no necessarily true.

Author Response

Reviewer 2

The authors propose a short review on hyper-excitbility in ALS. They decided to focus on cortical excitability only, missing the possible concomitant lower motor neuron hyperexcitability, clinically suggested by the presence of fasciculations early on the disease progression. In addition to this line, the authors wanted to defend the corticomotoneuronal hypotheis for ALS, in their contribution. This partial view did not consider the cons against this theory, which reduced the impact of their contribution. I would propose the authors should just keep their attention on the hyperexcitability topic.

1) A number of sentences are questionable. In introduction, indeed most axons present in the corticospinal tract have their origin outside the primary motor area, and the monosynaptic pathway is most related to hand discrete movements. In many familial cases no known genetic mutation is identified, but in about 10% sporadic case a genetic cause can be found.

Response

We thank the reviewer for these clarifications.  We acknowledge that the corticospinal tract has origins outside the primary motor area in both man and rodents and have added text and references to that effect; we have also added mention of the role of the monosynaptic pathway to hand movements (page 1, lines 32-34).  We have clarified the prevalence of genetic causes in both familial and sporadic ALS (page 1, line 45 through to page 2, line 60

2) Regarding Excitability section, spasticity and brisk reflexes derive from corticospinal tract lesion (like as stroke and spinal cord trauma) and not from cortical hyperexcitability.

Response

We thank our reviewer for the detailed comment on our paper. We have corrected the information regarding upper MN signs of spasticity and brisk reflexes not directly correlated with cortical hyper-excitability but as a consequence of corticospinal tract abnormality. (Page 2, line 82).

3) Really, cortical hyperexcitability causes epilepsy, which does not affect ALS patients. Both oculomotor nuclei and anal sphincter are (mildly) affected in ALS patients.

Response

We thank our reviewer for the feedback and have corrected the mild effect on both oculomotor nuclei and anal sphincter in ALS patients in our manuscript (Page 2, line 94).

4) Moreover, the clinical evidence for potassium channel dysfunction arrived from investigating peripheral axons (threshold tracking technique) and not from cortical studies (in humans).

Response

We thank and agree with our reviewer on the clinical evidence for potassium channel dysfunction arrived from investigating peripheral axons (threshold tracking technique) and not from cortical studies. We have therefore modified the information in our manuscript to avoid any misleading information (Page 4, line 161-165).

5) The way the authors approach these issues is misleading for readers. The same hold true for EAAT2 astroglial receptor dysfunction, demonstrated in the spinal cord from affected ALS patients (Rothstein et al).

Response

We thank our reviewer for pin-pointing on the details that were thought to be misleading to the readers. We have incorporated the specificity of the EAAT2 astroglial receptor dysfunction, demonstrated in the spinal cord from affected ALS patients (Rothstein et al) (Page 4, line 191).

Finally, early cortical hyperexcitability does not prove it antedates lower motor neuron changes. This is fashionable but no necessarily true.

Reviewer 3 Report

Manuscript ID: brainsci-1189988

This review describes the mechanism underlying hyperexcitability in amyotrophic lateral sclerosis (ALS).

Early cortical hyperexcitability was proposed to be one of the mechanisms leading to degeneration of motor neurons. This parameter was considered an interesting diagnostic marker in ALS.

The manuscript is well written and organized.

However there are some points that could be taken in consideration:

  • I suggest to give more details on the effects of drugs mentioned in the manuscript.
  • To improve the comprehension of the mechanisms I suggest to include 1 or 2 explaining figures.
  • Hyperexcitability has been observed also in skeletal muscle, due to different ion channel activity and expression. It should be interesting to mention that skeletal muscle hyperexcitability contribute to the pathology (Camerino et al., Sci Rep. 2019 Feb 28;9(1):3185. doi: 10.1038/s41598-019-39676-3.).

Minor points:

There are some mistakes:

Line 87: nucei

Line 97: Van den Bosh

Moreover, it can be better to number the bibliography?

Author Response

Reviewer 3

Manuscript ID: brainsci-1189988

This review describes the mechanism underlying hyperexcitability in amyotrophic lateral sclerosis (ALS).

Early cortical hyperexcitability was proposed to be one of the mechanisms leading to degeneration of motor neurons. This parameter was considered an interesting diagnostic marker in ALS.

The manuscript is well written and organized.

However there are some points that could be taken in consideration:

1) I suggest to give more details on the effects of drugs mentioned in the manuscript.

Response

We thank our reviewer for asking us to include more details on the effect of drugs mentioned in the manuscript. We have thus added the additional details about the drugs mentioned in the manuscript (Page 4, line 166; Page4, line 157).

2) To improve the comprehension of the mechanisms I suggest to include 1 or 2 explaining figures.

Response

3) Hyperexcitability has been observed also in skeletal muscle, due to different ion channel activity and expression. It should be interesting to mention that skeletal muscle hyperexcitability contribute to the pathology (Camerino et al., Sci Rep. 2019 Feb 28;9(1):3185. doi: 10.1038/s41598-019-39676-3.).

Response

We thank our reviewer for detailing the hallmark functional feature hyper-excitability observed across various cell types affected in ALS pathology. We have included the hyper-excitability observed in skeletal muscles and included the reference (Camerino et al., Sci Rep. 2019 Feb 28;9(1):3185. doi: 10.1038/s41598-019-39676-3). (Page 2, line 57-60).

Minor points:

There are some mistakes:

Line 87: nucei

Response: Changed accordingly to ‘nuclei’, (Page 2, line 93).

Line 97: Van den Bosh

Response: We thank the reviewer for pointing the details, however, we have confirmed that the author’s name was written as “van den Bos” as per pubmed citation reference (Page 3, line 103).

Round 2

Reviewer 2 Report

This reviewer is pleased with the current version